# Associations between Prenatal Exposure to Phthalates and Features of the Metabolic Syndrome in Males from Childhood into Adulthood

**DOI:** 10.3390/ijerph192215244

**Published:** 2022-11-18

**Authors:** Ye’elah E. Berman, Dorota A. Doherty, Trevor A. Mori, Lawrence J. Beilin, Oyekoya T. Ayonrinde, Leon A. Adams, Rae-Chi Huang, John K. Olynyk, Jeffrey A. Keelan, John P. Newnham, Roger J. Hart

**Affiliations:** 1Division of Obstetrics and Gynaecology, University of Western Australia, Perth, WA 6008, Australia; 2Medical School, University of Western Australia, Crawley, WA 6009, Australia; 3Department of Gastroenterology and Hepatology, Fiona Stanley Hospital, Murdoch, WA 6150, Australia; 4Department of Hepatology, Sir Charles Gairdner Hospital, Nedlands, WA 6009, Australia; 5School of Medical and Health Sciences, Edith Cowan University, Joondalup, WA 6027, Australia; 6Telethon Kids Institute, University of Western Australia, Nedlands, WA 6009, Australia

**Keywords:** boys, phthalate metabolites, adiposity, antenatal exposure, metabolic disorder, cardiometabolic

## Abstract

Phthalate metabolites are detectable within the majority of the population. Evidence suggests that a prenatal exposure to phthalates may be associated with the subsequent risks of obesity and elevated blood pressure. We hypothesised that a prenatal exposure to phthalates would lead to an increase in adverse cardiometabolic parameters through childhood and adulthood. The maternal serum phthalate measurements from the stored samples taken from Gen1 mothers at 18 and 34 weeks gestation were examined in relation to the cardiometabolic measures in 387 male offspring from the Raine Study. Data from the Gen2 follow-ups between 3 and 27 years were used. The primary outcomes were analysed longitudinally using linear mixed models for the repeated measures. Non-alcoholic fatty liver disease (NAFLD) was assessed at 17 years using logistic regression. A consistent positive relationship was observed between a prenatal exposure to mono-carboxy-iso-octyl phthalate (MCiOP) through adolescence into adulthood with systolic blood pressure. There were no other consistent cardiovascular associations. Mid-levels of prenatal exposures to Mono-n-butyl phthalate (MnBP) were associated with a greater incidence of NAFLD. Detectable Mono-3-carboxypropyl phthalate (MCPP) was associated with a lower serum HDL-C through late childhood into adulthood, while a higher prenatal exposure to mono-iso-butyl phthalate (MiBP), was associated with a higher LDL-C at 22 years of age. A mid-level prenatal exposure to mono-2-ethylhexyl phthalate (MEHP) metabolites was associated with higher insulin in adulthood, while a higher prenatal exposure to the sum of the Di-(2-ethyl-hexyl) phthalate (DEHP) and Di-iso-nonyl phthalate (DiNP) metabolites was associated with higher fasting serum glucose in adulthood. In conclusion, our study demonstrated that higher prenatal phthalate exposures to some phthalate metabolites was associated with some adverse metabolic profiles through adolescence into adulthood, although the consistent themes were limited to a few metabolites and the outcomes of systolic blood pressure, fasting insulin and glucose.

## 1. Introduction

The prevalence of overweight or obesity in individuals is increasing within populations [1]. Adiposity in childhood is a strong predictor of hypertension, dyslipidaemia, insulin resistance and the related adverse health outcomes in later life [2]. Hence, understanding the early life origins and predictors of obesity and metabolic disease is important to help reduce adverse health outcomes in later life.

There is an increasing awareness of the negative impact of environmental factors on human health. Evidence from animal models demonstrates that an exposure to endocrine-disrupting chemicals influences the offspring behaviour, metabolic risk, growth and reproduction [3,4,5]. Phthalates are ubiquitous chemicals that are widely used in industrial and consumer products, such as various plastics and personal care products. The absorption of phthalates occurs through oral, epidermal [6] and respiratory [7] routes (www.atsdr.cdc.gov: accessed on 11 December 2021). Furthermore, phthalates cross the placenta and may impact the developing foetus [8,9], with their metabolites detectable in amniotic fluid [10,11]. After absorption, phthalates are rapidly distributed via the circulatory system [12], being rapidly hydrolysed to monoesters. Additionally, a glucuronidation or oxidization of the more non-polar metabolites may occur [12]. Consequently, phthalate metabolites are detectable within the majority of the population at any one time [13,14], and an exposure to phthalate metabolites occurs throughout life [15]. Gaining a clear understanding of the influence of phthalate exposure on development and health is confounded by a number of factors including a non-linear dose response, their metabolic complexity, synergistic influences from other endocrine-disrupting chemicals [16], and their capacity to act with a low level of receptor occupancy [16]. The potential associations with adverse health outcomes are further compounded by the known associations with parental BMI, gestational smoking, diet in childhood and socio-economic status. 

Human and animal studies suggest that a prenatal exposure to phthalates is associated with a subsequent risk of obesity [5,16,17,18,19,20], the most consistent finding being a positive relationship between an exposure to di-ethyl phthalate (DEP) and BMI [21,22,23]. In addition, there is evidence that an early life exposure to phthalates is associated with adverse influences on systolic and diastolic blood pressure [19]. As a metabolic disturbance may be programmed early in utero [24], we hypothesised that a prenatal exposure to greater concentrations of phthalates would lead to an increase in cardiometabolic risk factors commencing in childhood and adolescence. The mechanisms of any potential adverse metabolic influence of phthalates may relate to their endocrine disrupting influence, or the fact that an exposure in the prenatal period has been demonstrated to be associated with adverse obstetric outcomes, such as preterm birth [25]. A prenatal exposure to phthalates has been associated with an increase in oxidative stress markers in pregnancy [5] but not with the development of gestational diabetes [18]. 

Using repeated cardiometabolic, anthropometric and DEXA measures from a well-established birth cohort, the Raine Study, we aimed to determine whether a maternal phthalate exposure was associated with the cardiometabolic risk factors and adiposity between birth and 20 years of age.

## 2. Materials and Methods

Data were sourced from the Raine Study (www.rainestudy.org.au: accessed on 11 December 2021), a longitudinal and multigenerational birth cohort study that investigated the safety and effects of ultrasound on the developing foetus. Between 1989 and 1992, 2900 pregnant women at 16 to 20 weeks gestation in Western Australia (WA) were recruited into the Study. These women delivered 2868 live born children [26]. Detailed anthropometric, respiratory, neurocognitive, endocrine, cardiovascular, physical fitness, metabolic, behavioural, psychological and social measurements were collected via surveys at ages 1, 2, 3, 5, 8, 10, 14, 17, 20, 22 and 27 years. Information on their maternal education and family income was collected prospectively. As the metabolic phenotype and outcomes differ between the sexes and may be influenced by oral contraceptive use in females, in this study analyses were performed on single males only. 

### 2.1. Management of Stored Maternal Blood Samples

Maternal serum was collected from the mothers (Gen1) at 18 and 34/36 weeks gestation and stored in aliquots without thawing at −80 °C. The samples from a thawed aliquot from both time points from 123 individuals were pooled for an analysis of the phthalate metabolites to provide an approximation of antenatal phthalate exposure across gestation. For each woman, 200 µL aliquots of the 18- and 34-week samples were pooled, frozen and couriered to Copenhagen, Denmark, for analysis. A previous pilot study confirmed the stability of the samples during prolonged storage at either −80 °C or up to 15 weeks at −20 °C and demonstrated an unaltered recovery when processed without an acid addition before storing or after storage [27].

### 2.2. Phthalate Measurements and Categorisation

The 982 maternal serum phthalate samples were analysed by isotope dilution LC-MS/MS with preceding enzymatic de-conjugation, as previously published [14]. The method for the preparation of the serum samples, standard solutions and quality controls, as well as the instrumental analysis and general method validation of 32 metabolites from 15 different phthalate diesters, has also been published [28]. Of the 982 available maternal serum phthalate samples, 437 were from women who were pregnant with a male foetus, and 387 of these Gen2 males had data available for at least one of the primary outcomes. The categorisation of the maternal serum phthalate levels was based on these 387 samples. Phthalate metabolites were included in the analysis if they had detectable levels in at least 10% of the 387 samples. This was the case for 12 of the phthalate metabolites and an additional 7 sums of the phthalate metabolites are all listed in Table 1. For phthalate metabolites where more than one third of the sample (N > 129) had undetectable levels, the phthalates were classified into a binary variable (detectable/undetectable). For the sums of phthalates, or where fewer than one third of the sample (*n* < 129) had undetectable phthalate levels, the phthalates were classified into tertiles as per the 33.3% and 66.6% cut points shown in the results tables. For each phthalate metabolite, and for the sums of the phthalate metabolites, the percent of the samples above the limit of detection (% > LOD), the minimum, median and maximum are shown in Appendix A.

### 2.3. Metabolic Assessments

#### 2.3.1. Hepatic Ultrasound

One hundred and eighty-six male participants in this study had both their maternal phthalates and hepatic ultrasound data at 17 years of age available. The methods used for the hepatic ultrasound have been reported previously, to determine the presence of non-alcoholic fatty liver disease (NAFLD) [29].

#### 2.3.2. Blood Pressure and Anthropometric Measurements

The systolic and diastolic blood pressure were measured at 3, 5, 8, 10, 14, 17, 20 and 22 years by trained nursing staff using an oscillometric sphygmomanometer (Dinamap vital signs monitor 8100, DINAMAP XL vital signs monitor and DINAMAP PROCARE 100 (DPC100X–EN)). Measurements were taken every two minutes for three readings to age ten, and for six readings in the subsequent follow-ups. The average of all the readings, excluding the first, were calculated and used for the analysis. Heart rate measurements were available at all but the 3-year follow-up. 

The height and weight were measured by Holtain Infantometer and Stadiometer to the nearest 0.1 cm (Holtain Ltd.) and Wedderburn Chair Scales (Wedderburn, Australia), or automatic electric scales, to the nearest 100 g thereafter. The body mass index (BMI) was calculated (kg/m^2^) and converted to z-scores using the Centers for Disease Control and Prevention (CDC) clinical growth charts [30]. The waist circumference, which was recorded twice at each of the 14, 17, 20 and 22-year follow-ups, was taken at the umbilicus to the closest 0.1 cm, and the average of the two measurements was used.

#### 2.3.3. Cardiometabolic Assessment

The blood concentrations of liver enzymes [alanine transaminase (ALT) and gamma-glutamyl transferase (GGT)], high density lipoprotein cholesterol (HDL-C), low density lipoprotein cholesterol (LDL-C), triglycerides (TG), high sensitivity CRP (hsCRP), glucose and insulin were measured at 14, 17, 20 and 22 years. The aspartate transaminase (AST) was measured at 17, 20 and 22 years and the uric acid and creatinine were measured at 14, 20 and 22 years. The blood samples were analysed at the PathWest Laboratory at Royal Perth Hospital, as described previously [31]. The homoeostatic model assessment (HOMA) was calculated as fasting the insulin concentrations (uU/mL) × fasting the glucose levels (mM/L)/22.5 [32].

Blood samples in which the participants were not fasting were excluded from the glucose, HOMA, triglycerides and LDL-C analyses. Participants with triglycerides > 4.5 mmol/L were removed from the LDL-C analysis. Diabetics (blood glucose > 7 mmol/L) were excluded from the insulin, glucose and HOMA analyses, and participants with hsCRP > 10 mg/L were removed from the hsCRP analysis.

#### 2.3.4. Dual Energy X-ray Absorptiometry

Dual energy X-ray absorptiometry (DEXA) was performed using a Norland XR-36 densitometer (Norland Medical Systems, Inc., Fort Atkinson, WI, USA) to assess the lean mass (g) and the fat mass (g) at 20 years of age. At the 27-year follow-up, the scan was conducted on the Lunar iDXA Advance Densitometer (GE Healthcare, Waukesha, WI, USA).

### 2.4. Statistical Analysis

The primary outcomes were the liver enzymes ALT, GGT and AST (U/L), systolic and diastolic blood pressure (mmHg), heart rate, fat mass (g), lean mass (g), non-alcoholic fatty liver disease (NAFLD), waist measurement (cm) and concentrations of HDL-C, LDL-C, TG, uric acid, hsCRP, glucose, insulin, HOMA and creatinine. The continuous data were summarised using medians and interquartile ranges with Mann–Whitney tests which were performed to compare the differences between the males included in the study (*n* = 387) and those who were excluded (*n* = 1012). Frequency distributions and chi square tests were used to compare the categorical outcomes. Logistic regression analyses were used to assess associations between the categorised phthalates and NAFLD at 17 years of age. The associations between the categorized phthalate metabolite levels and all the other primary endpoints were assessed longitudinally using linear mixed models for a repeated measures analysis. The marginal means for each outcome in its original units, by the maternal phthalate metabolite level, were reported at the average age of each follow-up for which that outcome was measured, and at the reference level of any included covariates. Differences in the slope over time, by the phthalate metabolite level, were reported for each outcome. Log transformations were performed on the skewed outcomes (waist circumference, ALT, GGT, AST, HDL-C, hsCRP, insulin, HOMA, triglycerides and fat mass), in which the case results were back-transformed to their original units and presented as marginal geometric means. Robust standard errors were used where the assumption of heteroscedasticity was violated or where the data were very skewed. Values of outcome variables below the limit of detection (LOD) were set to LOD/sqrt(2). All analyses were adjusted for the participants’ age at the measurement, with the analysis also adjusted for their household income at birth (<AUD7000, 7000–11,999, 12,000–23,999, 24,000–35,999, >36,000 or unknown), maternal education level at birth (<high school, high school, trade and tertiary degree), duration of breast feeding (<6 months, >6 months or unknown) and BMI z-score. The waist circumference, fat mass (g) and lean mass (g) were not adjusted for the BMI z-score, with the two later outcomes instead including an adjustment for the height.

Hypothesis tests were two sided with *p*-values of <0.05 considered to be statistically significant. As this was an exploratory, hypothesis-generating study of the associations between maternal phthalates and offspring’s metabolic outcomes to 27 years, no adjustments for multiple testing were made, as recommended by the American Statistical Association for such exploratory analyses [33,34]. SAS Enterprise Guide version 7.1 (SAS institute Inc., Cary, NC, USA) and STATA version 16 (StataCorp LLC., College Station, TX, USA) statistical software were used for the data analysis.

### 2.5. Ethical Approval

The study was conducted according to the guidelines of the Declaration of Helsinki and approved by the Raine Study Executive Committee and Human Research Ethics Committee of Princess Margaret Hospital. Adolescent participants and their accompanying parent or guardian provided informed written consent.

## 3. Results

There were 2868 live born offspring (Gen2) from which the study population was selected. Of the 1399 single males, 437 had maternal serum phthalate samples taken, and 387 of these also had one of the outcomes of interest available. The sample size included for each outcome is available in Figure 1. Compared to those not included, the Gen2 participants included in this study were slightly older at their 20-year follow-up, had higher levels of AST at age 20, had lower levels of GGT at 13 years of age, had a lower diastolic blood pressure at age 22, but had a higher systolic blood pressure at ages 3 and 5 (Appendix A). Their mothers were also slightly taller and less likely to have smoked during pregnancy but were otherwise similar (Table 2).

### 3.1. Systolic Blood Pressure

In the adjusted analysis, a significant positive association was detected between prenatal concentrations of mono(carboxyisooctyl) phthalate (MCiOP) and the systolic blood pressure between 10 and 22 years of age, with participants whose mothers had detectable levels of MCiOP having a higher systolic blood pressure than those with undetectable levels (Table 3 and Appendix A). A similar association was observed in the unadjusted analysis. In addition, in the unadjusted analysis, the participants whose mothers were in the middle tertile for mono-2-ethylhexyl phthalate (MEHP) had a lower systolic blood pressure at 3 and 5 years (Appendix A).

### 3.2. Diastolic Blood Pressure

In the adjusted analysis, a significant negative association was detected between prenatal mono(2-ethyl-5-carboxypentyl) phthalate (MECPP) concentrations and diastolic blood pressure between 10 and 20 years of age, with participants whose mothers had levels of MECPP in the middle tertile having lower levels of diastolic blood pressure than those in the lowest tertile. Additionally, participants whose mothers were in the highest tertile of antenatal monoethyl phthalate (MEP) during pregnancy had a significantly higher diastolic blood pressure at 3 years than those whose mothers were in the lowest tertile of MEP. Between 3 and 22 years of age, the usual incremental increase in diastolic blood pressure was attenuated in the group whose mothers were in the highest tertile for MEP compared with those in the lowest tertile (slope *p* = 0.007) (Table 3 and Appendix A). Similar associations were observed in the unadjusted analysis (Appendix A).

### 3.3. Heart Rate

In the adjusted analysis, a significant association was detected between prenatal monoisobutyl phthalate (MiBP) and heart rate between 5 and 14 years, and between prenatal mono-(3-carboxypropyl) phthalate (MCPP) and heart rate between 17 and 22 years. Gen2 participants whose mothers were in middle tertile MiBP levels during pregnancy had lower heart rates at 5, 8, 10 and 14 years of age compared to participants whose mothers were in the lowest tertile MiBP. Similarly, participants whose mothers had detectable levels of MCPP during pregnancy had a lower heart rate between 17 and 22 years than participants whose mothers had undetectable levels of MCPP. At age 10, participants whose mothers were in the middle tertile for mono-2-ethylhexyl phthalate (MEHP) during pregnancy had a higher average heart rate than participants whose mothers were in the lowest tertile (Table 3 and Appendix A). Similar associations were observed in the unadjusted analyses, with additional associations being observed between maternal MEP and Gen2 heart rate between 8 and 17 years, between maternal MECPP and Gen2 heart rate between 14 and 22 years and between maternal mono(2-carboxymethylhexyl) phthalate (MCMHP) and Gen2 heartrate at 20 and 22 years (Appendix A).

### 3.4. Waist Circumference

In the adjusted analysis, significant associations were observed between the waist circumference at ages 14 and 17 and maternal MEHP, and between the waist circumference at ages 14 and 17 and the maternal sum of di-(2-ethyl-hexyl) phthalate metabolites (∑DEHP), with participants whose mothers were in the middle tertile for MEHP or ∑DEHP having a lower waist circumference at 14 and 17 years than those in the lowest tertile MEHP or ∑DEHP. Additionally, participants whose mothers were in the middle tertile for MiBP had a larger waist circumference at 14 years than those in the lowest tertile MiBP (Table 3 and Appendix A). Similar results were observed in the unadjusted analyses (Appendix A).

### 3.5. Alanine Transaminase (ALT)

In the adjusted analysis, significant negative associations were observed between the maternal upper tertile MEHP and MECPP during pregnancy, and Gen2 ALT at 22 years. Gen2 participants whose mothers were in the upper tertile MEHP or MECPP during pregnancy had a lower ALT at 22 years than those whose mothers were in the lowest tertile MEHP or MECPP, and their ALT levels also increased significantly slower between 14 and 22 years (slope *p* = 0.002 and *p* = 0.005, respectively). The increase in the ALT levels were attenuated between 14 and 22 years for participants whose mothers had middle tertile MECPP or upper tertile MCMHP, compared to the lower tertiles of these phthalate metabolites (slope *p* = 0.026 and *p* = 0.048). Significant associations were observed between maternal middle tertile mono-isononyl phthalate (MiNP), the middle tertile sum of di-iso-nonyl phthalate metabolites (∑DiNP) or detectable MCiOP during pregnancy and Gen2 ALT at 14 years. Gen2 participants whose mothers had middle tertile MiNP or ∑DiNP, or detectable MCiOP during pregnancy, had a higher ALT at 14 years than those whose mothers had the lowest tertile MiNP or ∑DiNP, or undetectable MCiOP. For those whose mothers had middle tertile MiNP or ∑DiNP, their ALT levels increased more slowly between 14 and 22 years than those whose mothers had lower tertile MiNP or ∑DiNP (slope *p* = 0.018, *p* = 0.021, respectively). Finally, Gen2 participants whose mothers had upper tertile ∑DEHP had a higher ALT at 14 years, but a lower ALT at 22 years than participants whose mothers had lower tertile ∑DEHP during pregnancy. Participants whose mothers had upper or middle tertile ∑DEHP both had slower rises in their ALT between 14 and 22 years of age compared to participants whose mothers had lower tertile ∑DEHP (slope *p* = <0.001 and *p* = 0.020, respectively) (Table 3 and Appendix A). Similar associations were observed in the unadjusted analyses for MEHP, MCMHP, MiNP, ∑DiNP and ∑DEHP (Appendix A).

### 3.6. Gamma-Glutamyl Transferase (GGT) 

In the adjusted analysis, increases in the GGT between ages 14 and 22 were attenuated in Gen2 participants whose mothers had upper tertile MECPP, ∑DEHP or the sum of the high metabolic weight phthalate metabolites (∑highMWphth), compared to Gen2 participants whose mothers had low levels of these metabolites (slope *p* = 0.026, *p* = 0.049, *p* = 0.043, respectively). In addition, higher levels of GGT at age 14 were observed in Gen2 participants whose mothers had upper tertile ∑DiNP, compared to the lower tertile (Table 3 and Appendix A). In the unadjusted analyses, similar associations were observed for MEHP only. In addition, participants whose mothers had middle tertile MECPP during pregnancy had a significantly higher GGT at age 14 than participants whose mothers had lower tertile MECPP (Appendix A).

### 3.7. Aspartate Transaminase (AST)

In the adjusted analysis, Gen2 participants whose mothers had upper tertile MiNP or MECPP during pregnancy had a significantly lower AST than participants whose mothers had lower tertile MiNP or MECPP at 17 and 22 years, respectively. Participants whose mothers had upper tertile MECPP also had an attenuated increase in their AST between 17 and 22 years, compared to participants whose mothers had lower tertile MECPP (slope *p* = 0.041) (Table 3 and Appendix A). In the unadjusted analyses, similar associations were observed for MiNP only (Appendix A).

### 3.8. High Density Lipoprotein Cholesterol (HDL-C)

In the adjusted analysis, Gen2 participants whose mothers had a detectable MCPP during pregnancy had a lower HDL-C between 17 and 22 years compared to participants whose mothers had an undetectable MCPP during pregnancy. Furthermore, Gen2 participants whose mothers had upper tertile ∑DiNP during pregnancy had a significantly lower HDL-C at 14 years than participants whose mothers had lower tertile ∑DiNP during pregnancy (Table 3 and Appendix A). In the unadjusted analyses, the aforementioned associations were no longer significant, and there were associations between maternal MEHP during pregnancy and HDL-C between 17 and 22 years, maternal ∑DEHP during pregnancy and HDL-C at 17 and 20 years of age, and the maternal molar sum of Di-(2-ethyl-hexyl) phthalate and Di-iso-nonyl phthalate metabolites (∑DEHPDiNP) during pregnancy and HDL-C at 22 years (Appendix A).

### 3.9. Low Density Lipoprotein Cholesterol (LDL-C)

In the adjusted analysis, Gen2 participants whose mothers had upper tertile MiBP or the sum of the Di-iso-butyl phthalate and Di-n-butyl phthalate metabolites (∑MBP(i + *n*)), had significantly higher LDL cholesterol than those whose mothers had lower tertile MiBP or ∑MBP(i + *n*) at 22 years and between 20 and 22 years, respectively. Gen2 participants whose mothers had upper tertile ∑DEHP or middle tertile MiNP had a significantly lower LDL-C than those whose mothers had lower tertile ∑DEHP or MiNP at 22 years and from ages 17–22, respectively. Participants whose mothers had middle tertile MCMHP during pregnancy had a significantly greater increase in LDL-C between 14 and 22 years than participants whose mothers had lower tertile LDL-C during pregnancy (slope *p* = 0.049) (Table 3 and Appendix A). These associations were not significant in the unadjusted analyses (Appendix A).

### 3.10. Uric Acid

In the adjusted analysis, Gen2 participants whose mothers had middle tertile MiNP or ∑DiNP, or upper tertile MEP, during pregnancy had significantly lower uric acid at 14 years than participants whose mothers had lower tertile MiNP, ∑DiNP or MEP during pregnancy. Furthermore, participants whose mothers had middle tertile MiBP or ∑MBP (i + *n*) during pregnancy had significantly lower uric acid at 20 and 22 years than participants whose mothers had lower tertile MiBP or ∑MBP(i + *n*). Finally, participants whose mothers had middle tertile MiNP, the middle tertile ∑DINP or middle or upper tertile molar sum of all phthalate metabolites (∑allphth) had a greater increase in uric acid levels between 14 and 22 years than participants whose mothers were in the lowest tertiles of these phthalate metabolites (slope *p* = 0.026, *p* = 0.044, *p* = 0.039 and *p* = 0.047, respectively) (Table 3 and Appendix A). Similar associations were present in the unadjusted analyses for MEP, MiNP, ∑DiNP and ∑allphth (Appendix A).

### 3.11. High Sensitivity C-Reactive Protein (hsCRP)

In the adjusted analysis, Gen2 participants whose mothers had middle tertile ∑allphth or upper tertile MiBP during pregnancy had lower levels of hsCRP at 14 years and from 14 to 17 years, respectively, than participants whose mothers had lower tertile ∑allphth or MiBP. Participants whose mothers had upper tertile MiBP during pregnancy also had greater increases in their hsCRP from 14 to 22 years than participants whose mothers had lower tertile MiBP (slope *p* = 0.011). Gen2 participants whose mothers had upper tertile ∑DINP or middle or upper tertile MiNP during pregnancy had higher levels of hsCRP at 14 years and from 14 to 20 years, respectively, than participants whose mothers had lower tertile ∑DINP or MiNP. Participants whose mothers had upper tertile ∑DINP or MiNP also had a slower rise in hsCRP from 14 to 22 years, than participants whose mothers had lower tertile ∑DINP or MiNP during pregnancy (slope *p* = 0.014 and *p* = 0.010, respectively) (Table 3 and Appendix A). Similar associations were observed for MiBP, MiNP and ∑DiNP in the unadjusted analyses (Appendix A). 

### 3.12. Insulin

In the adjusted analysis, Gen2 participants whose mothers had middle tertile MEHP during pregnancy had higher levels of insulin at the 20- and 22-years mark than participants whose mothers had lower tertile MEHP during pregnancy. The insulin levels of participants whose mothers had middle or upper tertile MEHP during pregnancy also decreased more slowly between 14 and 22 years than participants whose mothers had lower tertile MEHP (slope *p* = 0.050 and *p* = 0.032, respectively) (Table 3 and Appendix A). In the unadjusted analyses, there was no significant association between the maternal serum MEHP level during pregnancy and the insulin levels between ages 14 and 22. However, there was an association between maternal serum MCMHP during pregnancy and Gen2 insulin levels at age 14 (Appendix A). 

### 3.13. Glucose

In the adjusted analysis, Gen2 participants whose mothers had upper tertile MiBP during pregnancy had lower levels of glucose at 14 and 17 years than participants whose mothers had lower tertile MiBP during pregnancy. Gen2 participants whose mothers had upper tertile ∑DEHPDiNP or detectable mono-benzyl phthalate (MBzP) during pregnancy had higher levels of glucose at the 22-years mark than participants whose mothers had lower tertile ∑DEHPDiNP or undetectable MBzP during pregnancy. The glucose levels also rose faster between 14 and 22 years in participants whose mothers had detectable MBzP than those whose mothers had undetectable MBzP (slope *p* = 0.007) (Table 3 and Appendix A). In the unadjusted analyses, similar associations were found between glucose and maternal MiBP and MBzP (Appendix A).

### 3.14. Homeostatic Model Assessment (HOMA)

In the adjusted analysis, Gen2 participants whose mothers had middle tertile MEHP during pregnancy had a significantly higher HOMA at 20 and 22 years than participants whose mothers had lower tertile MEHP during pregnancy. The decrease in the HOMA was attenuated between 14 and 22 years in participants whose mothers had upper tertile MEHP compared to participants whose mothers had lower tertile MEHP (slope *p* = 0.036) (Table 3 and Appendix A). In the unadjusted analyses, there was no significant association between the MEHP and HOMA, but participants whose mothers had upper tertile MCMHP during pregnancy had a significantly lower HOMA at age 14 than participants whose mothers had lower tertile MCMHP during pregnancy (Appendix A).

### 3.15. Creatinine

In the adjusted analysis, Gen2 participants whose mothers had upper tertile MECPP or MCMHP during pregnancy had a higher creatinine at 20 and 22 years than participants whose mothers had lower tertile MECPP or MCMHP. Gen2 participants whose mothers had middle tertile ∑DEHP during pregnancy also had a higher creatinine at 22 years than participants whose mothers had a lower tertile ∑DEHP. Participants whose mothers had upper tertile MCMHP or middle tertile ∑DEHP during pregnancy also had a greater increase in creatinine between ages 14 and 22 compared to participants whose mothers had lower tertile MCMHP or ∑DEHP during pregnancy (slope *p* = 0.027 and *p* = 0.006, respectively) (Table 3 and Appendix A). Similar associations were observed in the unadjusted analyses (Appendix A).

### 3.16. Triglycerides (TG)

In the adjusted analysis, Gen2 participants whose mothers had middle tertile mono-n-butyl phthalate (MnBP) or ∑MBP(i + *n*) during pregnancy had higher TG from 17 to 22 years of age than participants whose mothers had lower tertile MnBP or ∑MBP(i + *n*) during pregnancy. Gen2 participants whose mothers had middle tertile ∑DiNP or upper tertile ∑DEHPDiNP had a lower TG at 17 years than participants whose mothers had lower tertile DiNP or ∑DEHPDiNP, and participants whose mothers had upper tertile ∑DEHP had a lower TG at 14 and 17 years than participants whose mothers had lower tertile ∑DEHP. Finally, participants whose mothers had upper tertile MEHP or middle tertile MiNP during pregnancy had a lower TG from 14 to 20 years and 17 to 20 years, respectively, than participants whose mothers had lower tertile MEHP or MiNP during pregnancy (Table 3 and Appendix A). In the unadjusted analyses, similar associations were observed for MnBP, MEHP, ∑MBP(i + *n*), ∑DEHP and ∑DEHPDiNP. Additional associations were observed between maternal MECPP, mono-iso-decyl phthalate (MiDP) and ∑highMWphth during pregnancy, with Gen2 TG during adolescence (Appendix A).

### 3.17. Dual Energy X-ray Absorptiometry (DEXA)-Fat Mass

In the adjusted analysis, Gen2 participants whose mothers had middle tertile MECPP had a significantly greater increase in fat mass between 20 and 27 years than participants whose mothers had lower tertile MECPP (Table 3 and Appendix A). The same association was observed in the unadjusted analysis (Appendix A). 

### 3.18. Dual Energy X-ray Absorptiometry (DEXA)-Lean Mass

In the adjusted analysis, Gen2 participants whose mothers had middle or upper tertile MEHP, middle tertile ∑DiNP, upper tertile ∑DEHP, ∑DEHPDiNP or ∑highMWphth gained significantly more lean mass between 20 and 27 years than participants whose mothers were in the lowest tertile for concentrations of these phthalate metabolites during pregnancy. Participants whose mothers had the middle tertile sum of low molecular weight phthalate metabolites (∑low MWphth) during pregnancy gained less lean mass between 20 and 27 years than participants whose mothers had lowest tertile ∑low MWphth during pregnancy. Finally, participants whose mothers had detectable MCiOP or upper tertile MECPP or MCMHP during pregnancy had a significantly higher lean mass at age 27 than participants whose mothers had undetectable MCiOP or lower tertile MECPP or MCMHP during pregnancy (Table 3 and Appendix A). In the unadjusted analyses, similar associations were observed for MEHP, MCiOP, ∑DEHP, ∑DiNP, ∑DEHPDiNP, ∑low MWphth and ∑highMWphth. In addition, participants whose mothers had upper tertile ∑DiNP, ∑DEHPDiNP or ∑highMWphth had a higher lean mass at 27 years than participants whose mothers were in the lowest tertile of these phthalate metabolites. Associations between a change in the lean mass between 20 and 27 years and maternal ∑allphth and MiNP were also observed in the unadjusted analysis (Appendix A). 

### 3.19. Hepatic Ultrasound-Non-Alcoholic Fatty Liver Disease (NAFLD)

In the adjusted analysis, Gen2 participants whose mothers had middle tertile MnBP during pregnancy had a significantly higher risk of NAFLD than participants whose mothers had lower tertile MnBP (Table 3 and Appendix A). A similar association was observed in the unadjusted analysis (Appendix A).

## 4. Discussion

This longitudinal study of the prospective association between maternal phthalate metabolite exposures during pregnancy and cardiometabolic measures through childhood into adulthood in males demonstrated mixed findings, with a potential association of an adverse metabolic profile with a higher prenatal exposure to some phthalate metabolites. 

The only notable cardiovascular associations were a prenatal MCiOP exposure and systolic blood pressure through adolescence into early adulthood, and a prenatal MEP exposure with higher diastolic blood pressure at 3 years of age. At 10 years of age, a higher resting heart rate was associated with a prenatal exposure to MEHP. 

With respect to the hepatic outcome measures, higher prenatal MEHP and MECPP metabolites were associated with lower concentrations of a serum ALT concentration into adulthood, although, there were no consistent associations of a prenatal phthalate exposure and the ALT, AST and GGT concentrations. Regarding the presence of NAFLD at 17 years of age, mid-levels of prenatal exposures to MnBP were associated with a greater incidence of NAFLD than lower levels of prenatal exposures, but no other significant findings were detected. With these inconsistent findings, it is unlikely that there exists an association between a prenatal phthalate exposure and hepatic dysfunction.

Interestingly, there appeared to be an association of an adverse lipid profile subsequent to a prenatal phthalate metabolite exposure, as detectable MCPP was associated with lower serum HDL-C during adolescence and adulthood, and higher sums of DiNP metabolites during pregnancy were associated with lower serum HDL-C concentrations at 14 years. Furthermore, a higher prenatal exposure to MiBP and the sum of the metabolites of MBP (i + *n*)) were associated with a higher LDL-C in early adulthood. In contrast, the sum of the DEHP metabolites were associated with lower LDL-C concentrations in early adulthood. No consistent associations between a prenatal phthalate exposure and serum concentrations of triglycerides during childhood into adolescence were detected. Higher fasting serum insulin concentrations in adolescence were associated with a mid-level prenatal exposure to MEHP metabolites, and consequently a higher HOMA score in adolescence and adulthood. In a similar vein, a higher prenatal exposure to the sum of the DEHP and DiNP metabolites was associated with higher fasted serum glucose concentrations in adulthood. Consequently, there is a potential theme of some phthalate metabolites being associated with an adverse lipid profile.

With respect to a marker of inflammation, hsCRP, higher prenatal concentrations of MiNP were associated with higher concentrations of hsCRP in adolescence, but there were no consistent associations between a prenatal phthalate exposure and serum concentrations of uric acid through childhood into adulthood. Higher prenatal MECPP and MCMHP exposures were associated with higher concentrations of creatinine in adulthood, and a greater rate of rise in creatinine during adolescence than low exposure levels.

Only male data are presented here as sexually dimorphic influences of prenatal exposures to phthalates on the growth parameters have been reported [35] and to avoid the complexity of any influence on the metabolic parameters by the use of oral contraceptive pills. 

With respect to adiposity, the most consistent finding reported in the literature is an association with a DEP exposure. Data derived from the study by Center for the Health Assessment of Mothers and Children of Salinas (CHAMACOS) reported that urinary metabolites of DEP, butyl benzyl phthalate (BBzP) and DEHP were positively associated with the BMI z-score, waist circumference z-score and body fat in young girls [22], and an exposure to DEHP metabolites was associated with an increased BMI across the ages [21], similar to that reported by Yang et al. [23]. Whilst confirming the positive association of the BMI trajectory through childhood with the sum of the metabolites of DEHP and DiNP, Kupsco et al. reported that higher MCNP exposures were associated with a lower BMI trajectory through childhood [20]. We have previously reported that prenatal phthalate metabolite exposures were not associated with any consistent influences on the measures of female and male adiposity [36,37], as measured by the body mass index and dual energy X-ray absorpitometry (DEXA), although our data suggest that mid-level prenatal exposures to the metabolites of MECPP are associated with a greater gain in fat mass between 20 and 27 years [21,22,23]. It also appeared that low level exposures to the low molecular weight and higher exposures to the higher molecular weight were associated with a faster lean mass accrual between 20 and 27 years. 

The literature pertaining to early life exposures to phthalates with cardiometabolic outcomes in children is summarised by the meta-analysis by Golestanzadeh et al. who reported a positive association between both low and high molecular weight phthalates and BMI through childhood [19]. Furthermore, they reported positive associations of exposure to phthalates and systolic and diastolic blood pressure in childhood, but no consistent associations between different phthalates and waist circumference, HDL-C and TG, and they concluded by stating that society should be making attempts to reduce our reliance on our usage of phthalates. With respect to the liver metabolism, a mouse model of a prenatal exposure to low-dose DEHP has shown the subsequent development of the metabolic syndrome, with an abnormal adipogenesis, an altered glucose metabolism and an altered hepatic thiamine metabolism [17].

Whilst our analysis may have derived a potential association of prenatal exposures to phthalates with some adverse trends in the metabolic parameters, it did not derive any consistent associations between prenatal phthalate exposures and any anthropometric, hepatic or cardiovascular parameters through childhood into adulthood, thus it is important to state that the interpretation of the data can be complicated due to some phthalates having differential effects at different stages of development and a non-linear dose–response for some phthalates. This is further compounded by the difficulty of controlling for the known influences of parental BMI, gestational smoking, diet in childhood, socio-economic status and phthalate exposure in adolescence, which appears to increase the abdominal fat deposition [38]. Furthermore, small numbers and missing data prevented an adjustment for some additional antenatal (gestational diet and alcohol intake) and postnatal (diet, smoking and alcohol intake in adolescence) exposures associated with the metabolic outcomes. 

We cannot exclude that our findings are caused by multiple statistical testing. It also would have increased our study power considerably to have urine samples from pregnancy as a matrix for an exposure assessment, as the phthalate concentrations in urine are higher than in serum, hence, more samples lie within the detectable range. This would increase the sensitivity to find linear associations. Furthermore, the potential for contamination during sampling would be reduced [14]. We performed a series of pilot studies to ensure that the results of the phthalate analysis were robust and, therefore, were likely to reflect a phthalate exposure at the time of the collection [27] and addressed these concerns in previous publications [28]. In addition, to support the veracity of our measures, the phthalate metabolites levels observed in our study are consistent with those reported in previous studies [39,40,41].

It is important to point out that any associations derived from these analyses of an antenatal phthalate exposure cannot confirm a direct causal relation to the outcome. This is a cohort study, and our findings are of an epidemiological nature. Further research is required to verify whether the same effects are observed in vivo. Furthermore, the magnitude of the changes reported were small, and hence the biological significance may be limited, or even non-existent. We were limited in our ability to control for the potential for the influence of additional postnatal exposures to phthalates, or other endocrine-disrupting chemicals, or lifestyle factors known to affect metabolic indices during pregnancy or the cohort follow-up. Further, as phthalates are non-persistent chemicals, the exposure to them can vary from day to day, although individuals tend to follow their exposure trajectories [42]. Consequently, despite the potential for the limitations of their collection and storage, the phthalate metabolite concentrations that were measured are likely to reflect an in vivo exposure at the time of the collection. 

## 5. Conclusions

While we detected some possible associations of a prenatal phthalate exposure with a subsequent increase in systolic blood pressure and altered lipid and glucose profiles, our long-term study of prenatal phthalate exposures on cardiovascular, hepatic and metabolic indices through childhood and into adulthood did not demonstrate any consistent associations. The data did suggest that a phthalate exposure may be associated with the subsequent development of an adverse lipid profile, and an MCiOP or MEP exposure may be positively associated with systolic and diastolic blood pressure, respectively.

## Figures and Tables

**Figure 1 ijerph-19-15244-f001:**
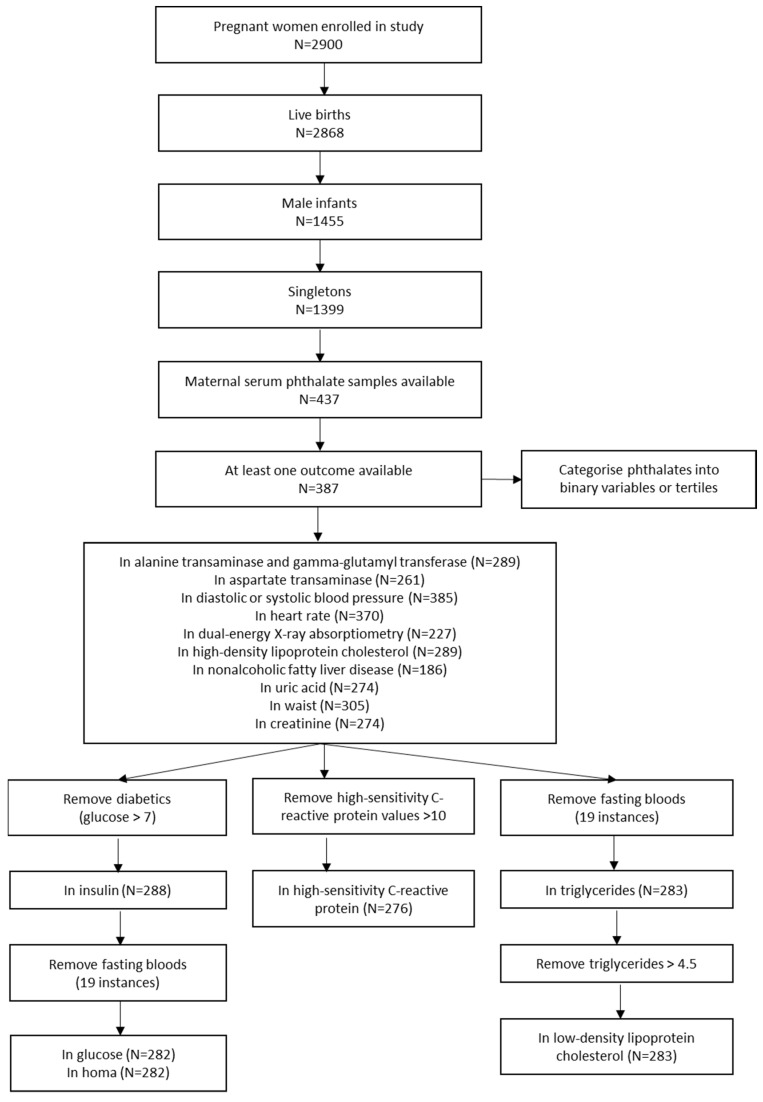
Flow Chart of Study Participants.

**Table 1 ijerph-19-15244-t001:** Phthalate diesters and their respective metabolites detected in maternal serum during pregnancy and sums of metabolites used for statistical analysis.

Phthalate Diester		Human Serum Metabolite	
Di-ethyl phthalate	DEP	Mono-ethyl phthalate	MEP
Di-iso-butyl phthalate	DiBP	Mono-iso-butyl phthalate	MiBP
Di-n-butyl phthalate	DnBP	Mono-n-butyl phthalate	MnBP
		Mono-(3-hydroxybutyl) phthalate	MHBP ^a^
Butylbenzyl phthalate	BBzP	Mono-benzyl phthalate	MBzP
Di-(2-ethyl-hexyl) phthalate	DEHP	Mono-(2-ethyl-hexyl) phthalate	MEHP
		Mono-(2-ethyl-5-carboxypentyl) phthalate	MECPP ^a^
		Mono-(2-carboxymethyl-hexyl) phthalate	MCMHP ^a^
Di-octyl phthalate	DOP	Mono-3-carboxypropyl phthalate	MCPP ^a^
Di-iso-nonyl phthalate	DiNP	Mono-iso-nonyl phthalate	MiNP
		Mono-carboxy-iso-octyl phthalate	MCiOP ^a^
Di-iso-decyl phthalate	DiDP	Mono-iso-decyl phthalate	MiDP
**Sums of phthalate metabolites**			
ΣMBP(i + *n*)		Sum of MiBP and MnBP in ng/mL	
∑DEHPmetab		Molar sum of MEHP, MCMHP, and MECPP expressed as DEHP in ng/mL	
∑DiNPmetab		Molar sum of MiNP and MCIOP expressed as DiNP in ng/mL	
∑DEHPmetab + ∑DiNPmetab		Molar sum of MEHP, MCMHP, MECPP, MiNP, MCiOP expressed as MEHP in ng/mL	
∑low MW phth.metab		Molar sum of MEP, MiBP, MnBP, and MHBP expressed as MEP in ng/mL	
∑high MW phth.metab		Molar sum of MBzP, MEHP, MCMHP, MECPP, MCPP, MiNP, MCiOP, and MiDP expressed as MEHP in ng/mL	
∑all phth.metab		Molar sum of MEP, MiBP, MnBP, MHBP, MBzP, MEHP, MCMHP, MECPP, MCPP, MiNP, MCiOP, and MiDP expressed as MEHP in ng/mL	

^a^ Secondary metabolite, all other metabolites are primary metabolites.

**Table 2 ijerph-19-15244-t002:** Characteristics of the single males in the Raine Study who were (*n* = 387) and were not (*n* = 1012) included in the present analysis.

	Not Included in Study		Included in Study		
	(Maternal Serum Phthalate Levels or Outcome Data Unavailable) *n* = 1012		(Maternal Serum Phthalate Levels and Outcome Data Available) *n* = 387		
	N	Median (IQR) or N(%)	N	Median (IQR) or N(%)	*p*
**Maternal characteristics in pregnancy**					
Maternal age at delivery (years)	1009	27.9 (23.5, 31.8)	386	28.9 (24.3, 32.5)	0.084
Smoked during pregnancy	1012	270 (26.7)	387	79 (20.4)	**0.015**
Maternal height (m)	1012	1.6 (1.6, 1.7)	387	1.6 (1.6, 1.7)	**0.038**
Maternal BMI category	1012		387		0.629
Underweight		114 (11.3)		42 (10.9)	
Normal		710 (70.3)		282 (72.9)	
Overweight		118 (11.7)		43 (11.1)	
Obese		70 (6.9)		20 (5.2)	
Maternal education since leaving school	1012		387		0.969
<High school		331 (32.7)		124 (32.0)	
High school		201 (19.9)		77 (19.9)	
Trade		384 (37.9)		146 (37.7)	
University		96 (9.5)		40 (10.3)	
Family Income at birth	957		371		0.609
<$7000		90 (9.4)		31 (8.36)	
$7000–$11,999		88 (9.2)		32 (8.63)	
$12,000–$23,999		246 (25.7)		86 (23.2)	
$24,000–$35,999		237 (24.8)		91 (24.5)	
>$36,000		296 (30.9)		131 (35.3)	
Breastfeeding of Gen2 participant	827		359		0.3318
Never breastfed		87 (10.5)		28 (7.8)	
breastfed for <6 months		308 (37.2)		141 (39.3)	
breastfed for >6 months		432 (52.2)		190 (52.9)	
**Participants characteristics at birth**					
Gestational age at birth	1012		387		
<34 weeks		32 (3.2)		7 (1.8)	
34–36 weeks		69 (6.8)		18 (4.7)	
37+ weeks		911 (90.0)		362 (93.5)	
Birthweight (kg)	1012	3.4 (3.1, 3.8)	387	3.5 (3.1, 3.8)	0.738

**Table 3 ijerph-19-15244-t003:** Summary of all significant adjusted associations between phthalate metabolites/sums of phthalates, and each outcome, by follow-up age.

	Follow-Up								
Outcome	3 Years	5 Years	8 Years	10 Years	14 Years	17 Years	20 Years	22 Years	27 Years
Systolic blood pressure				MCiOP	MCiOP	MCiOP	MCiOP	MCiOP	
Diastolic blood pressure	High MEP								
				Mid MECPP	Mid MECPP	Mid MECPP	Mid MECPP		
Heart rate		Mid MiBP	Mid MiBP	Mid MiBP	Mid MiBP				
				Mid MEHP					
						MCPP	MCPP	MCPP	
Waist circumference					Mid MiBP				
					Mid MEHP	Mid MEHP			
					Mid ∑DEHP	Mid ∑DEHP			
Fat mass									
Lean mass									High MECPP
									High MCMHP
									MCiOP
Alanine transaminase								High MEHP	
								High MECPP	
					Mid MiNP				
					MCiOP				
					High ∑DEHP			High ∑DEHP	
					Mid ∑DiNP				
Gamma-glutamyl transferase					High ∑DiNP				
Aspartate transaminase								High MECPP	
						High MiNP			
Non-alcoholic fatty liver disease						Mid MnBP			
High-density lipoprotein cholesterol						MCPP	MCPP	MCPP	
					High ∑DiNP				
Low-density lipoprotein cholesterol								High MiBP	
						Mid MiNP	Mid MiNP	Mid MiNP	
							High ∑MBP(i + *n*)	High ∑MBP(i + *n*)	
								High ∑DEHP	
Triglycerides						Mid MnBP	Mid MnBP	Mid MnBP	
					High MEHP	High MEHP	High MEHP		
						Mid MiNP	Mid MiNP		
						Mid ∑MBP(i + *n*)	Mid ∑MBP(i + *n*)	Mid ∑MBP(i + *n*)	
					High ∑DEHP	High ∑DEHP			
						Mid ∑DINP			
						High ∑DEHPDINP			
Uric acid					High MEP				
							Mid MiBP	Mid MiBP	
					Mid MiNP				
							Mid ∑MBP(i + *n*)	Mid ∑MBP(i + *n*)	
					Mid ∑DINP				
High-sensitivity C-reactive protein					High MiBP	High MiBP			
					Mid MiNP	Mid MiNP	Mid MiNP		
					High MiNP				
					High ∑DiNP				
					Mid ∑allphth				
Insulin							Mid MEHP	Mid MEHP	
Homeostatic model assessment							Mid MEHP	Mid MEHP	
Glucose					High MiBP	High MiBP			
								MbZP	
								High ∑DEHPDINP	
Creatinine							High MECPP	High MECPP	
							High MCMHP	High MCMHP	
								Mid ∑DEHP	

1. ‘Mid’ and ‘High’ indicate that the middle or upper tertile, respectively, were significantly different from the lowest tertile. Where the tertile is not specified, the metabolite has been binarized and refers to detectable levels, with non-detectable levels of the metabolite as the reference. 2. Red text indicates a directly proportional relationship. As phthalate metabolite levels go up, the outcome also increases. 3. Green text indicates an inversely proportional relationship. As phthalate metabolite levels go up, the outcome decreases. 4. Grey boxes indicate that data were not collected for that outcome, at that follow-up age.

## Data Availability

The data underlying this article were provided by the Raine Study by permission. Data will be shared on request to the corresponding author with permission of The Raine Study.

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
