# Peer review of "Associations between Prenatal Exposure to Phthalates and Features of the Metabolic Syndrome in Males from Childhood into Adulthood"

_ijerph, 2022, doi:10.3390/ijerph192215244_

Round 1

Reviewer 1 Report

Interesting topicand sound presentation. 

One suggestion which may improve the clarity and readability of the paper. The authors may wish to include a few sentences and refs to justify and highlight the importance of their research objective. Where in the literature is the need to further investigate the hypothesis that ..."phthalates would lead to an increase in adverse cardiometabolic parameters..." grounded?

Author Response

Thank you for your comments. Please see the following text on page 2, lines 70-77, providing the need to further investigate this hypothesis: ‘Human and animal studies suggest that prenatal exposure to phthalates is associated with subsequent risk of obesity [5,16-20], the most consistent finding being a positive relationship between of di-ethyl phthalate (DEP) exposure and BMI [21-23]. In addition, there is evidence that early life exposure to phthalates is associated with adverse influences on systolic and diastolic blood pressure [19]. As metabolic disturbance may be programmed early in-utero [24], we hypothesised that prenatal exposure to greater concentrations of phthalates would lead to an increase in cardiometabolic risk factors commencing in childhood and adolescence.’

Reviewer 2 Report

The study reviewed herein examined the association between prenatal phthalate exposure, as assessed by maternal phthalate serum levels at 18 and 34 weeks gestation, and subsequent offspring metabolic health metrics at multiple timepoints after birth. The overall design and conduct of the study appear appropriate. However, there are a few issues that should be clarified prior to publication of the article:

1) This reviewer notes that logistical regression was performed on phthalate measures in tertiles but not maternal phthalate levels analyzed as a continuous variable. The authors should either perform logistical regression on non-categorized phthalate levels or provide a justification as to why such an analysis would be inappropriate.

Author Response

Thank you for your comment. We chose to analyse the independent variable, phthalate level, as a categorical variable rather than as a continuous variable due to the distribution of the maternal phthalate levels. Firstly, for each phthalate metabolite, a lot of mothers had levels below the limit of detection, making it very difficult to analyse as a continuous variable. Secondly, even without these values below the limit of detection, the distributions were all extremely positively skewed to the point that we couldn’t find any transformation to normalise them. For these reasons we decided to categorise maternal phthalate levels, which also meant that this method of analysis was consistent with our previous papers on the subject:

  • Berman YE, Doherty DA, Main KM, Frederiksen H, Hickey M, Keelan JA, Newnham JP, Hart RJ. Associations between prenatal exposure to phthalates and timing of menarche and growth and adiposity into adulthood: a twenty-years birth cohort study. International journal of environmental research and public health. 2021 Jan;18(9):4725.
  • Ye'elah EB, Doherty DA, Main KM, Frederiksen H, Keelan JA, Newnham JP, Hart RJ. The influence of prenatal exposure to phthalates on subsequent male growth and body composition in adolescence. Environmental research. 2021 Apr 1;195:110313.

Reviewer 3 Report

This study, by Dr. Berman et al, demonstrate that higher prenatal phthalate exposures to some phthalate metabolites was associated with some adverse metabolic profile through adolescence into adulthood. However, even if there is a correlation, the causal relation is unknown. Besides, phthalate and its metabolites are not the only cause of metabolic syndrome, but combined effects with other chemical substances are not considered. Although this study is an important finding in epidemiology, it is necessary to verify whether prenatal phthalate exposures induce adverse cardiometabolic parameters in vivo.

Author Response

Thank you for your comments. We agree that any correlations that we detected do not infer causation, and have stated this on page 17, line 564 ‘It is important to point out that any associations derived from these analyses of antenatal phthalate exposure cannot confirm a direct causal relation to outcome’.

Unfortunately, the data we have does not allow us to assess the combined effects of phthalate metabolites with other chemical substances. On line 569 we acknowledged that ‘We were limited in our ability to control for the potential for the influence of additional postnatal exposures to phthalates, or other endocrine disrupting chemicals, or lifestyle factors known to affect metabolic indices during the cohort follow-up’.  We have now amended this sentence to reflect that we also could not control for the influence of these factors during pregnancy.

To address the reviewer’s last point, we have added a sentence at page 17, line 565 stating ‘This is a cohort study, and our findings are of an epidemiological nature. Further research is required to verify whether the same effects are observed in vivo.’

Reviewer 4 Report

In general,

The manuscript entitled " Associations between Prenatal Exposure to Phthalates and Features of the Metabolic Syndrome in Males from Childhood into Adulthood" is an interesting characterization of that prenatal exposure to phthalates would lead to an increase in adverse cardiometabolic parameters through childhood and adulthood.

The Instrument is well described, the Materials and Methods are clear and the Conclusions are valuable and useful for many in the field interested in the analysis.

However, it would be valuable to address the following comments:

In the abstract include the definition of the abbreviation MEHP.

Correct the °C in all the writing where it applies, this because it appears written in different ways in the writing.

Author Response

Thank you these comments. Please see the definition for the abbreviation for MEHP at page 1, line 34 and the corrected OC on page 3, lines 101, 106 and 107.

Round 2

Reviewer 3 Report

A clear answer is given to the comment.